# The Use of Artificial Intelligence for Estimating Anterior Chamber Depth from Slit-Lamp Images Developed Using Anterior-Segment Optical Coherence Tomography

**DOI:** 10.3390/bioengineering11101005

**Published:** 2024-10-09

**Authors:** Eisuke Shimizu, Kenta Tanaka, Hiroki Nishimura, Naomichi Agata, Makoto Tanji, Shintato Nakayama, Rohan Jeetendra Khemlani, Ryota Yokoiwa, Shinri Sato, Daisuke Shiba, Yasunori Sato

**Affiliations:** 1OUI Inc., Tokyo 107-0062, Japan; 2Yokohama Keiai Eye Clinic, Kanagawa 240-0065, Japan; 3Department of Ophthalmology, Keio University School of Medicine, Tokyo 160-8582, Japan; 4Department of Biostatistics, Keio University School of Medicine, Tokyo 160-8582, Japan

**Keywords:** artificial intelligence, anterior chamber depth, machine learning, slit-lamp images, anterior-segment optical coherence tomography, Smart Eye Camera, glaucoma, deep learning, algorithm, telemedicine

## Abstract

Primary angle closure glaucoma (PACG) is a major cause of visual impairment, particularly in Asia. Although effective screening tools are necessary, the current gold standard is complex and time-consuming, requiring extensive expertise. Artificial intelligence has introduced new opportunities for innovation in ophthalmic imaging. Anterior chamber depth (ACD) is a key risk factor for angle closure and has been suggested as a quick screening parameter for PACG. This study aims to develop an AI algorithm to quantitatively predict ACD from anterior segment photographs captured using a portable smartphone slit-lamp microscope. We retrospectively collected 204,639 frames from 1586 eyes, with ACD values obtained by anterior-segment OCT. We developed two models, (Model 1) diagnosable frame extraction and (Model 2) ACD estimation, using SWSL ResNet as the machine learning model. Model 1 achieved an accuracy of 0.994. Model 2 achieved an MAE of 0.093 ± 0.082 mm, an MSE of 0.123 ± 0.170 mm, and a correlation of R = 0.953. Furthermore, our model’s estimation of the risk for angle closure showed a sensitivity of 0.943, specificity of 0.902, and an area under the curve (AUC) of 0.923 (95%CI: 0.878–0.968). We successfully developed a high-performance ACD estimation model, laying the groundwork for predicting other quantitative measurements relevant to PACG screening.

## 1. Introduction

Primary angle closure glaucoma (PACG) significantly contributes to visual impairment, particularly in Asia, where detection rates are notably low [1]. One of the challenges in managing PACG is its often asymptomatic nature in the early stages, making early detection crucial to prevent vision loss [2]. Therefore, the community-based detection and monitoring of angle closure are essential, especially in regions with high prevalence rates like Asia, where up to 64.7% of PACG cases are undetected [1].

Despite the necessity for effective screening tools, the current gold standard, gonioscopy, presents several limitations. It is a complex, time-consuming procedure requiring significant technical expertise, access to a slit-lamp biomicroscope, and the application of local anesthesia [3].

Recent advancements in artificial intelligence (AI) have opened new avenues for innovation in ophthalmic imaging. AI has been successfully applied in image analysis for disease screening, receiving regulatory approval for retinal disease screening [4,5,6,7,8,9,10,11,12,13,14]. Although fewer AI algorithms exist for anterior segment eye diseases, studies indicate their potential in effectively detecting conditions such as pterygium from anterior segment optical coherence tomography (AS-OCT) scans [15].

Anterior chamber depth (ACD) is a significant risk factor for angle closure and has been suggested as a quick screening parameter for primary angle closure disease [16] and as a tool for PACG screening [16,17]. However, the availability of ocular biometers for ACD measurements is limited in primary care or community settings because the biometers are stationary, large, and expensive, restricting their use for widespread screening [16,17].

Given these challenges, this study aims to develop an AI algorithm to predict ACD quantitatively from anterior segment photographs captured using a portable smartphone-based slit-lamp microscope similar to our past studies [18,19,20]. This approach leverages the portability and accessibility of smartphones, offering a practical solution for community-based screenings where advanced imaging tools are not readily available. By validating this algorithm, we aim to establish PACD screening, ultimately enhancing early detection and intervention efforts in high-risk populations.

## 2. Materials and Methods

### 2.1. Ethics Approval

In accordance with the principles of the Declaration of Helsinki, this study was conducted following the protocols approved by the Institutional Ethics Review Board of the Minamiaoyama Eye Clinic, Tokyo, Japan (IRB No. 202101). Given the retrospective nature of the study and the utilization of deidentified data, the need for written informed consent was exempted.

### 2.2. Study Design

In this retrospective analysis, data were sourced from the Yokohama Keiai Eye Clinic, a single ophthalmology institution. All videos recorded between July 2020 and December 2021 were collated. The videos were captured using the Smart Eye Camera (SEC; SLM-i07/SLM-i08SE, OUI Inc., Tokyo, Japan; 13B2X10198030101/13B2X10198030201). The ophthalmologists consistently employed the standard direct focal illumination method, using a full-height slit-beam of 0.1 to 0.3 mm (mm) thickness, angled at 40°. The slit-lamp videos needed to display a well-focused frame of both the cornea and crystalline lens for a minimum of five seconds each, as illustrated in Figure 1. The inclusion criteria for the videos were: (1) eyes in non-mydriatic conditions; (2) a clear focus on the cornea and crystalline lens, enabling the observation of the anterior chamber depth (ACD); (3) a minimum video duration of five seconds. Conversely, videos were excluded for reasons such as difficulty in integration into the study. Specific exclusion criteria for patients encompassed: (1) corneal ailments impeding ACD assessment (e.g., bullous keratopathy and corneal opacity); (2) aphakic and pseudophakic eyes or/and mydriasis eyes; (3) videos of subpar quality, notably those devoid of any crystalline lens frame. Following these criteria, 15 eyes of 6 cases were excluded, and a dataset of 1586 eyes was curated for the dataset. Parallelly, ACD measurements were derived to all of the subjects by AS-OCT utilizing the CASIA2 Advance system (Tomey, Nagoya, Japan). All assessments were executed under consistent lighting conditions in a designated room.

For the development of the ML model, videos and ACD values were collated on a cloud server and organized into a dataset explicitly designed for this investigation. During the preprocessing stage, videos were decomposed into static images, resulting in the extraction of 204,639 anterior segment images from videos encompassing 1586 eyes from 797 cases. Subsequently, these frames were annotated by ophthalmologists (E.S. and S.S.) to determine whether they could be classified as “diagnosable,” which accurately recognized ACD, or “non-diagnosable”, where ACD recognition was inconclusive. A ML algorithm was then employed to segregate the frames into these categories as an initial step in our analysis. Following this, all 204,639 frames were reassessed for “diagnosable” or “non-diagnosable” status, with subsequent steps only incorporating frames deemed “diagnosable” (as detailed in the Datasets and Machine Learning section, Model 1). These selected frames were then divided into training, validation, and test datasets at proportions of 80%, 10%, and 10%, respectively, in preparation for the subsequent ML phase and validated the model. In this second ML phase which estimates ACD (Model 2), training involves the use of ACD values obtained from AS-OCT in tandem with anterior segment images. Finally, the model underwent validation and testing phases to assess its performance (Figure 1).

### 2.3. Portable Recording Slit-Light Microscope

SEC was employed as a diagnostic tool to record slit-light videos. Designed as a smartphone attachment, the SEC has showcased comparable diagnostic capabilities to traditional slit-lamp microscopes, as evidenced in animal studies [21] and various clinical studies [22,23,24,25]. It emulates traditional diagnostic techniques such as slit-lamp microscope, particularly in the diagnosis of ACD in clinical settings [22]. The SEC projects a slit light with a thickness ranging from 0.1 to 0.3 mm, tailored to visualize the crystalline lens within both nondilated and dilated pupils at a consistent angle of 40° [22]. Given its capacity to capture anterior segment videos of the eyes, the SEC was selected to maximize video recordings and subsequently aggregate a substantial dataset of cataract images. The devices utilized for recording were the iPhone 7 and iPhone SE2 (Apple Inc., Cupertino, CA, USA). Videos were recorded with resolutions spanning from 720 × 1280 to 1080 × 1920 pixels, maintaining frame rates of either 30 or 60 frames per second.

### 2.4. Anterior Segment Optical Coherence Tomography

AS-OCT measurements were conducted using the CASIA2 Advance system. Certified optometrists examined all participants, ensuring the acquisition of a minimum of two sets of high-quality images. All OCT scans were pupil-centered and taken along the vertical axis, capturing the superior and inferior angles at 90° and 270°, respectively, in alignment with the CASIA system’s standard anterior-segment scanning protocol [22]. ACD was quantified as the vertical distance between the posterior corneal margin’s apex (specifically, the corneal endothelium) and the apex of the crystalline lens, as delineated by AS-OCT [22]. To ensure data quality, certified optometrist and a medical doctor (H.N., and R.J.K) meticulously reviewed the AS-OCT data to secure two sets of pristine images per patient. This step ensured the automated software’s proficiency in digitizing and identifying the corneal and lens surfaces, aligning with the study’s objective criteria.

### 2.5. Datasets and Machine Learning (ML)

Initially, data stored on our cloud server was migrated to a local server to facilitate ML analysis. The videos were decomposed into static images, with each frame being classified by multiple ophthalmologists (E.S. and S.S.) as either “diagnosable” or “non-diagnosable”. “Diagnosable” frames were identified as those providing a clear view of the anterior chamber, including the cornea, anterior chamber, and crystalline lens, suitable for estimating ACD. Conversely, “non-diagnosable” frames were defined as those not meeting the specified criteria for diagnosable images. In this first ML phase (Model 1), an ML algorithm was utilized to automatically categorize anterior segment images using the SWSL ResNet model [26]. Following this, all 204,639 frames were evaluated and labeled as either “diagnosable” or “non-diagnosable”, with the latter being excluded from further analysis. The remaining diagnosable frames were then randomly allocated into training, validation, and test datasets. Specifically, 1252 eyes, corresponding to 29,397 frames, were assigned to the training dataset. Meanwhile, 159 eyes with 3855 frames, and 175 eyes with 4027 frames were allocated to the validation and test datasets, respectively. Machine learning processing was exclusively applied to the training dataset, with the other datasets reserved for validation purposes. To determine ACD values through our ML model, a deep learning algorithm was trained to predict ACDs using normalized images of the anterior segment as input (Model 2). The SWSL ResNet model was applied to the training data for this purpose [26]. For the visualization of class activation mapping, we implemented the gradient-weighted class activation mapping (Grad-CAM) method [27,28,29]. Grad-CAM is a sophisticated post hoc visual explanation technique that highlights the areas within an image that are pivotal for a deep neural network’s predictions by visualizing the gradient of the class score as it relates to the input image. The computational setup for this experiment included a CPU of Intel^®^ Core i9-13900H (Intel Corporation, Santa Clara, CA, USA) with 20 cores, paired with a GPU of NVIDIA^®^ GeForce RTX™ 4090 Laptop (NVIDIA Corporation, Santa Clara, CA, USA).

### 2.6. Statistical Analysis

Given that this is a pioneering study, determining an ideal sample size was challenging. As such, we enrolled as many cases as possible to ensure robust results. Significance tests were conducted based on their respective confidence intervals. To assess the performance of the machine learning-based ACD estimation model, we utilized two metrics, the mean absolute error (MAE) and mean squared error (MSE), which measure the accuracy of a model’s predictions. We calculated the MAE and MSE based on both individual frame inputs and the cumulative input from entire video sequences for each eye. Spearman’s correlation coefficients were determined for the correlation analysis between values by the AI algorithm and values of AS-OCT. The accuracy, sensitivity, specificity, and area under the receiver operating characteristic curve (AUC) were calculated based on the ACD values. The values were classified as deep or shallow using cutoff ratios of 2.400 mm or 3.000 mm. A 2 × 2 contingency table was then created to compute these metrics. All statistical analyses were performed using SPSS (ver. 29; International Business Machines Corporation, Armonk, NY, USA).

## 3. Results

### 3.1. Demographics of the Datasets

In our study, we used a dataset consisting of 204,639 frames obtained from 1586 eyes of 797 cases from the dataset included individuals aged between 18 and 97 years. Additionally, the dataset encompassed a single ethnic composition, including Asian populations (Figure 1). These frames were pivotal for the training, validation, and testing of our machine learning models. Regarding the distribution of ACD within the dataset: 16 eyes had an ACD of less than 2.000 mm, 158 eyes fell within the ACD range of 2.000–2.500 mm, 538 eyes in the 2.500–3.000 mm range, 712 eyes in the 3.000–3.500 mm range, 176 eyes in the 3.500–4.000 mm range, and only 1 eye exhibited an ACD of over 4.000 mm (range: 1.070–5.030 mm; average ± standard deviation [SD]: 3.018 ± 0.385 mm; measured by AS-OCT, as shown in Figure 2).

### 3.2. Performance of the Diagnosable Frame Extraction Model (Result of Model 1)

Model 1 was developed with the dual functionality of extracting a diagnostically relevant frame representing the ACD from the video data and determining whether the frame pertains to the left eye or right eye. This model demonstrated exceptional performance metrics in its operational capacity, achieving an accuracy of 0.994, a precision of 0.995, and a recall of 0.995 (Figure 3A). Furthermore, when specifically assessing the model’s capability to accurately classify frames as either pertaining to the left eye or right eye, it attained an accuracy of 0.939 (Figure 3A).

### 3.3. Performance of the ACD Estimation Model (Result of Model 2)

Model 2 is an AI model designed to estimate the ACD from the diagnosable frames extracted by Model 1. Our AI algorithm demonstrated capability in estimating the ACD with a MAE of 0.118 ± 0.099 mm and a MSE of 0.154 ± 0.200 mm on a per-frame basis. Figure 4B). Additionally, upon evaluation across individual eyes, the estimation accuracy of the AI algorithm improved, yielding an MAE of 0.064 ± 0.071 mm and an MSE of 0.096 ± 0.148 mm, further illustrating the model‘s enhanced precision in ACD estimation on a per-eye basis (Figure 3B). Moreover, the performance of Model 2 achieved an MAE of 0.093 ± 0.082 mm and an MSE of 0.123 ± 0.170 mm on a per-case (Figure 3B).

### 3.4. Correlation of the Estimated Values versus AS-OCT Values

Within the scope of our investigation, the analysis of the relationship between the actual ACD measurements obtained via AS-OCT and the estimations provided by our predictive model for individual frames demonstrated a statistically significant strong correlation (R = 0.928, 95% confidence interval [CI] [0.924–0.932], Spearman’s rank correlation, as shown in Figure 4A). Similarly, the evaluation of this correlation across all eyes revealed a comparably significant strong association between AS-OCT-measured ACD values and those estimated by our model (R = 0.963, 95% CI [0.951–0.972], Spearman’s rank correlation coefficient, as shown in Figure 4B). Moreover, a similarly strong correlation was observed between the AS-OCT-measured ACD values and those estimated by our model (R = 0.953, 95% CI [0.912–0.961], Spearman’s rank correlation coefficient; Figure 4C).

### 3.5. Estimation of the Risk for Angle Closure Glaucoma

In estimating the risk of angle closure glaucoma, our model demonstrated high accuracy, sensitivity, specificity, and AUC with an ACD cut-off value of 2.400 mm (accuracy: 0.983, 95% CI [0.960–0.983]; sensitivity: 1.000, 95% CI [0.987–1.000]; specificity: 0.727, 95% CI [0.533–0.727]; AUC: 0.729, 95% CI [0.596–0.863]; Figure 5A). Conversely, when the ACD cut-off value was set to 3.000 mm, the model exhibited high accuracy, sensitivity, specificity, and AUC (accuracy: 0.922, 95% CI [0.874–0.951]; sensitivity: 0.943, 95% CI [0.894–0.972]; specificity: 0.902, 95% CI [0.855–0.930]; AUC: 0.923, 95% CI [0.878–0.968]; Figure 5B).

### 3.6. Visualization

In the Grad-CAM visualization, the overlaid heatmap on the input images indicates that the area with the highest intensity corresponds to the middle of the anterior chamber (Figure 6).

## 4. Discussion

This study evaluated the capability of our machine learning model to estimate the ACD from videos recorded using SEC. ACD measurements obtained via AS-OCT were used as the gold standard. The model‘s efficacy was quantified using MAE, MSE, correlation, and the estimated risk of angle closure based on ACD. The ACD distribution in our cohort was first assessed for diversity and representativeness. Previous studies report mean ACD values ranging from 2.79 mm to 3.26 mm [30,31]. A histogram of our cohort‘s ACD distribution closely resembled that of a larger cohort of approximately 5000 individuals [32], supporting the representativeness of our sample.

To assess the sufficiency of our developed model compared to past studies, it is important to note that limited research has been carried out previously. Soh, Z.D. et al. used 2311 pairs of anterior segment photos and ocular biometer data from AS-OCT, achieving an MAE of 0.18 ± 0.14 mm in open angles and 0.19 ± 0.14 mm in angle closure, with an R² of 0.63 in validation data [33]. Chen, D. et al. used a portable slit-lamp prototype with smartphone-corrected images from 66 eyes, achieving an R^2^ of 0.73 [34]. Qian, C. et al. utilized 4157 smartphone-based anterior segment photos, achieving an MAE of 0.16 ± 0.13 mm and an R^2^ of 0.40 [35]. In our current study, we achieved higher performance with an MAE of 0.093 ± 0.082 mm, an MSE of 0.123 ± 0.170 mm, and an R of 0.953 between predicted and measured ACD (Figure 3 and Figure 4). Regarding angle closure risk assessment using AI, Qian, Z. et al. achieved 80% sensitivity, 79% specificity, and an AUC of 0.86 with 3753 anterior segment photos [36]. Our model achieved a sensitivity of 0.943 and a specificity of 0.902 with an ACD cutoff of 3.000 mm, and a sensitivity of 1.000, specificity of 0.727, and AUC of 0.991 with an ACD cutoff of 2.400 mm (Figure 5), demonstrating better performance with both 3.000 mm and 2.400 mm cutoff compared to previous studies. The success of our model can be attributed to (1) the larger dataset size (total 37,279 frames) [37], (2) the use of a single type of portable slit-lamp device (SEC) for consistent image correction, (3) the application of video recording to increase dataset numbers [18,19,20], and (4) the fixed slit angle of 40 degrees in the SEC, which may contribute to image stabilization [18,19]. These factors contribute to the higher performance of our model compared to previous studies.

This study has several limitations. First, the dataset is limited, as it includes data only from a single institute and exclusively from phakic eyes without mydriasis. Physiological changes in ACD after dilation are significant due to the posterior movement of the crystalline lens caused by the dilated iris–lens diaphragm, leading to changes in ACD values compared to initial measurements [38]. Additionally, data from eyes with intraocular lenses (IOL) or aphakic eyes were not collected. ACD in eyes with IOL tends to be deeper due to the thickness and position of the IOL [39]. Devereux et al. reported that using a screening cutoff of <2.22 mm could effectively distinguish primary angle closure from normal eyes [40], and recent studies have reinforced these findings, demonstrating higher sensitivity and specificity using the same cutoff value, thereby supporting the utility of ACD measurements in screening for angle closure conditions [1]. Our study used cutoffs of 2.400 mm and 3.000 mm due to the limited diversity in the collected datasets (Figure 2). To apply our model in a real clinical setting, it is necessary to collect more diverse data, including mydriatic eyes, eyes with IOL, aphakic eyes, and a wider range of ACD values from multiple institutes.

Secondly, we utilized only the SEC for data collection. Previous studies have shown that images of the anterior segment captured with the SEC are of sufficient quality to evaluate various anterior segment diseases, including estimating ACD [21,22], not only in Japanese patients but also in Indian, Indonesian, and Italian populations [23,24,25]. However, the quality of anterior segment images can vary depending on the brand of slit-lamp microscopes used. Therefore, further studies are required to evaluate images taken with different slit-lamp microscopes, as demonstrated by Ueno, Y. et al. [41]. Moreover, we could not reproduce conventional diagnosis method such as gonioscopy, so future studies are needed to gain more precise diagnosis methods.

Thirdly, we employed the SWSL ResNet as our machine learning model based on its proven efficiency in handling medical imaging data, particularly when annotated data are limited [26]. SWSL ResNet is a semi-supervised learning (SSL) technique that leverages a large collection of unlabeled images to enhance the performance of state-of-the-art image classification methods. In a similar study, Soh, Z.D. et al. utilized ResNet-50, an architecture that applies residual learning with labeled images [27]. Likewise, Qian Z employed ResNet-34 as a machine learning model [36]. Other algorithms, such as the estimation of tear meniscus height from keratography images, have utilized U-net as a central neural network model, achieving an accuracy of 82.5%, sensitivity of 0.899, precision of 0.911, and F1 score of 0.901 from 217 images [42]. Elsawy, A et al. developed a deep learning neural network diagnosis algorithm for three anterior segment diseases using VGG19, achieving an AUC of 0.94 to 0.99 from AS-OCT images [43]. These previous studies, including those on ACD estimation and other anterior segment diseases, have predominantly used labeled images as resources for machine learning. The use of labeled images is common in AI algorithm development for the anterior segment of the eyes [44]. In contrast, our study applies an SSL model that utilizes unlabeled images, which may be one of the reasons our model achieves high performance. However, since most past studies have used labeled images, future studies will need to explore different machine learning models for further improvement.

Despite the limitations, our AI algorithm offers a promising tool for estimating ACD from slit-lamp images. To the best of our knowledge, no other tools currently exist that can record anterior segment videos and apply an AI algorithm to estimate ACD in a portable format. Therefore, this tool could be instrumental in risk assessment for acute glaucoma attacks, particularly in developing countries and rural clinics where access to sophisticated ophthalmological equipment is limited combined with telemedicine [2,45,46,47,48,49]. Moreover, including more data about family history, data of intraocular pressure, refraction, and other parameters, we are able to screen the primary angle closure glaucoma. Future studies are warranted to validate our model in larger and more diverse populations and to explore its integration into standard clinical workflows.

## 5. Patents

E.S. is a founder of OUI Inc. OUI Inc. has the patent for the Smart Eye Camera (Patent No. JP; 6627071, USA; 16/964822, EU; 19743494.7, China; 201980010174.7, India; 202017033428, VN; 1-2020-04893, and Africa; AP/P2020/012569. Patent pending EU; 2175926.2, US; 17/799043). There are no other relevant declarations relating to this patent.

## Figures and Tables

**Figure 1 bioengineering-11-01005-f001:**
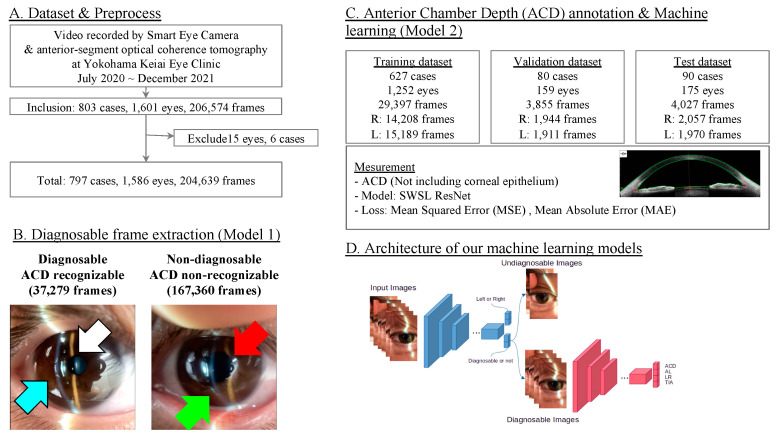
Study design and model developments. (**A**) Dataset and preprocessing of study chart. (**B**) Diagnosable frame extraction (Model 1). Images classified as diagnosable and non-diagnosable. (**C**) Anterior chamber depth annotation and machine learning (Model 2). Dataset is split into training, validation, and test datasets. (**D**) Architecture of our machine learning models.

**Figure 2 bioengineering-11-01005-f002:**
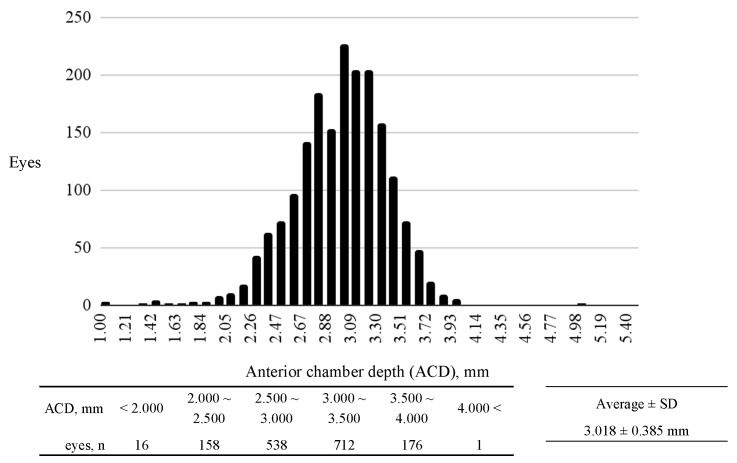
Distribution of anterior chamber depth (ACD) in our study. The x-axis represents the ACD in millimeters (mm), ranging from 1.800 mm to 4.000 mm, divided into 0.100 mm increments. The y-axis indicates the number of eyes corresponding to each ACD measurement range. The average ACD across the study population is 3.018 ± 0.385 mm.

**Figure 3 bioengineering-11-01005-f003:**
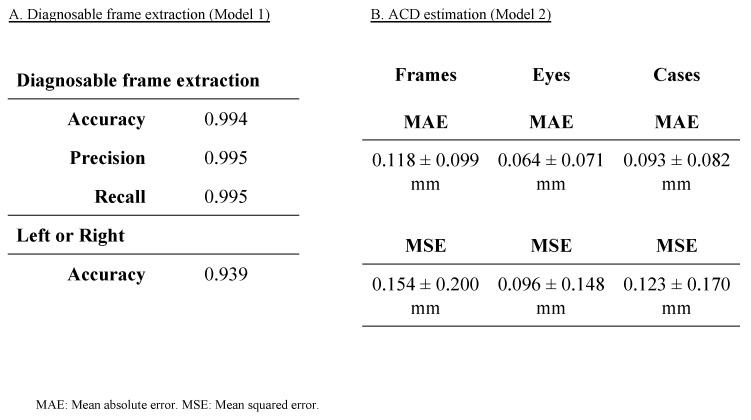
Performance metrics for diagnosable frame extraction (Model 1) and anterior chamber depth (ACD) estimation (Model 2). Figure 3 presents the performance metrics for the two models used in the study. Model 1 focuses on extracting diagnosable frames. Model 2 evaluates the accuracy of ACD estimation.

**Figure 4 bioengineering-11-01005-f004:**
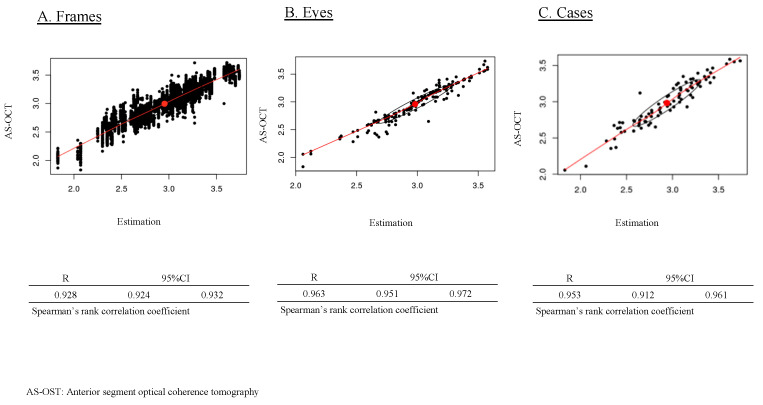
Correlation between AS-OCT anterior chamber depth (ACD) values and AI-estimated ACD measurements. Figure 4 illustrates the correlation between AS-OCT-measured ACD and the AI-estimated ACD from both individual frames and eyes.

**Figure 5 bioengineering-11-01005-f005:**
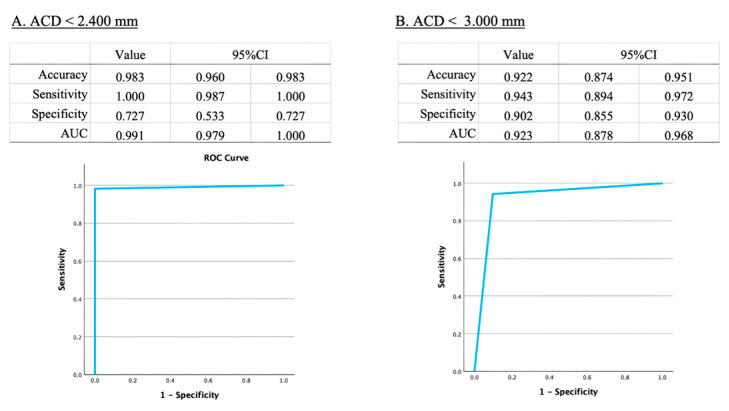
Diagnostic performance of anterior chamber depth (ACD) cut-off values for angle closure glaucoma. Figure 5 highlights the comparison of diagnostic performance metrics for different ACD cut-off values in terms of sensitivity, specificity, accuracy, and area under the curve (AUC).

**Figure 6 bioengineering-11-01005-f006:**
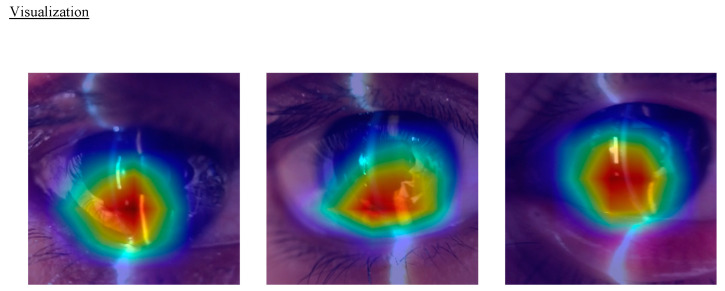
Visualizations of anterior chamber depth (ACD) estimation model. These visualizations provide insights into the areas of the eye that are most influential in the model’s ACD estimation process using gradient-weighted class activation mapping (Grad-CAM) method. It shows that the heatmap images pointing out where the anterior chamber is.

## Data Availability

The data used to support this study’s findings are available by contacting the corresponding author upon reasonable request.

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
