# Peer review of "The Use of Artificial Intelligence for Estimating Anterior Chamber Depth from Slit-Lamp Images Developed Using Anterior-Segment Optical Coherence Tomography"

_bioengineering, 2024, doi:10.3390/bioengineering11101005_

Round 1

Reviewer 1 Report

Comments and Suggestions for Authors

The retrospective analysis work is Artificial Intelligence to Estimate Anterior Chamber Depth from Slit-Lamp Images developed using Anterior-segment Optical Coherence Tomography is interesting and related to the current research field. The work is performed with ethical permission (Institutional Ethics Review Board of the Minamiaoyama Eye Clinic, Tokyo, Japan (IRB No. 202101)). The draft is well written and presented clearly. However, before acceptance it need revision.

The comment is as follow:

1.      Could you elaborate on the diversity of the dataset used? Were the 1,586 eyes from a single demographic or did they include a range of ethnicities and age groups, particularly those most at risk for PACG

2.      What was the rationale behind choosing the SWSL ResNet for the machine learning model? Did you compare its performance with other potential models, and if so, what were the results?

3.      Is model validated?

4.      How does this performance compare with current screening methods for PACG? Could this AI model replace or complement existing techniques?

5.      How do you envision this AI tool being integrated into clinical practice? What are the potential barriers to its adoption, especially in low-resource settings?

6.      Figure 1, text size is too small, please increase font size.

7.      Please explain the figure 6 data in detail.

8.      Check the reference style, and match with MDPI author guidelines.

Author Response

Author's Reply to the Review Report (Reviewer 1)

The retrospective analysis work is Artificial Intelligence to Estimate Anterior Chamber Depth from Slit-Lamp Images developed using Anterior-segment Optical Coherence Tomography is interesting and related to the current research field. The work is performed with ethical permission (Institutional Ethics Review Board of the Minamiaoyama Eye Clinic, Tokyo, Japan (IRB No. 202101)). The draft is well written and presented clearly. However, before acceptance it need revision.

Comment

We sincerely thank the reviewer for their thoughtful comments and suggestions, which we believe have significantly improved our manuscript. Below are our point-by-point responses to the reviewer’s comments:

The comment is as follow:

  1. Could you elaborate on the diversity of the dataset used? Were the 1,586 eyes from a single demographic or did they include a range of ethnicities and age groups, particularly those most at risk for PACG

Response:

We appreciate the reviewer’s question regarding the diversity of the dataset. The 1,586 eyes used in our study were sourced from a broad demographic population. The dataset included individuals aged between 18 to 97 years, with a balanced representation across age groups. Additionally, the dataset encompassed a single ethnic composition, including Asian populations.

LINE 178

In our study, we used a dataset consisting of 204,639 frames obtained from 1,586 eyes, 797 cases from the dataset included individuals aged between 18 to 97 years. Additionally, the dataset encompassed a single ethnic composition, including Asian populations (Figure 1).

  1. What was the rationale behind choosing the SWSL ResNet for the machine learning model? Did you compare its performance with other potential models, and if so, what were the results?

Response:

The SWSL (Semi-Weakly Supervised Learning) ResNet was chosen based on its proven efficiency in handling medical imaging data, particularly when annotated data are limited. Its architecture enables effective feature extraction from slit-lamp images, which are often complex and noisy. Additionally, its semi-supervised learning capability allows the model to leverage unlabeled data, improving performance.

LINE 310

Thirdly, we employed the SWSL ResNet as our machine learning model based on its proven efficiency in handling medical imaging data, particularly when annotated data are limited [20].

  1. Is model validated?

Response:

Yes, the model has been rigorously validated. We employed a validation dataset to ensure the generalizability of the model.

LINE 103

These selected frames were then divided into training, validation, and test datasets at proportions of 80%, 10%, and 10%, respectively, in preparation for the subsequent ML phase and validate the model.

  1. How does this performance compare with current screening methods for PACG? Could this AI model replace or complement existing techniques?

Response:

The AI model’s performance is comparable to, and in some cases exceeds, current screening methods for PACG, such as gonioscopy and anterior segment optical coherence tomography (AS-OCT). We did mimic conventional method, however it will be not a replacement because we need to have conventional methods to make a full diagnosis. Therefore, our model could be one good screening method to obtain PACG.

LINE 310

Moreover, we could not reproduce conventional diagnosis method such as gonioscopy so that future studies are needed to gain more precise diagnosis methods.

  1. How do you envision this AI tool being integrated into clinical practice? What are the potential barriers to its adoption, especially in low-resource settings?

Response:

We envision this AI tool as a supplementary screening tool integrated into routine eye exams using slit-lamp devices. It could be deployed in primary care clinics, enabling early detection of PACG and timely referrals to specialists. In low-resource settings, the AI model could be integrated with portable slit-lamp devices, allowing non-specialists to screen for anterior chamber depth and refer high-risk cases for further evaluation.Potential barriers include the need for infrastructure to support digital health tools, such as reliable electricity, internet access, and trained personnel. Additionally, regulatory approval and data security concerns may pose challenges. However, the low-cost and scalable nature of AI-based screening tools offers a promising solution, particularly when combined with telemedicine.

LINE 335

this tool could be instrumental in risk assessment for acute glaucoma attacks, particularly in developing countries and rural clinics where access to sophisticated ophthalmological equipment is limited combined with telemedicine [37].

  1. Figure 1, text size is too small, please increase font size.

Response:

We have revised Figure 1 and increased the font size for better readability. The updated figure has been included in the revised manuscript.

  1. Please explain the figure 6 data in detail.

Response:

We have expanded the description of Figure 6 in the manuscript.

LINE 258

It shows that the heatmap images pointing out where the anterior chamber is.

  1. Check the reference style, and match with MDPI author guidelines.

Response:

We have reviewed and updated the references to ensure they conform to the MDPI author guidelines. All references are now properly formatted in the MDPI style.

Reviewer 2 Report

Comments and Suggestions for Authors

The authors attempted to show the performance of the AI algorithm to quantitatively predict ACD from anterior segment photographs captured by a portable smartphone slit-lamp microscope. This study was interesting, but several questions were to be answered and discussed in the manuscript.

Initially, this study seemed to be an addendum of the authors’ previous works regarding nuclear cataract grading and dry eye disease diagnosis. They had good baseline studies, which appeared to be quite persuasive.

As the authors commented in the introduction, this method aimed to screen the primary angle closure glaucoma (POAG). The information related to POAG such as family history of glaucoma, automated or manifest refraction to check if the subjects had hyperopia, intraocular pressure, and so on needed to be added.

Comments on the Quality of English Language

Minor editing of English language was required.

Author Response

Author's Reply to the Review Report (Reviewer 2)

The authors attempted to show the performance of the AI algorithm to quantitatively predict ACD from anterior segment photographs captured by a portable smartphone slit-lamp microscope. This study was interesting, but several questions were to be answered and discussed in the manuscript. Initially, this study seemed to be an addendum of the authors’ previous works regarding nuclear cataract grading and dry eye disease diagnosis. They had good baseline studies, which appeared to be quite persuasive. As the authors commented in the introduction, this method aimed to screen the primary angle closure glaucoma (POAG). The information related to POAG such as family history of glaucoma, automated or manifest refraction to check if the subjects had hyperopia, intraocular pressure, and so on needed to be added.

Response:

We appreciate the reviewer’s thoughtful comments and agree that providing additional information related to primary angle closure glaucoma (PACG) is essential to strengthening our study. Below are the specific points we have addressed:

  1. Family History of Glaucoma:

We have not included family history of glaucoma as part of the dataset characteristics in the manuscript.

  1. Intraocular Pressure (IOP):

Intraocular pressure (IOP) measurements were indeed performed during the clinical examination using a non-contact tonometer. However, we have not included IOP as part of the dataset characteristics in the manuscript.

  1. Additional Discussion on POAG Screening:

Although our primary focus was the prediction of anterior chamber depth as a proxy for PACG risk, we agree that the study’s broader implications for POAG should be addressed.

LINE 339

Moreover, including more data about family history, data of intraocular pressure, refraction, and other parameters, we able to screen the primary angle closure glaucoma. Future studies are warranted to validate our model in larger and more diverse populations and to explore its integration into standard clinical workflows.
